# CSP: An Efficient Baseline for Learning on Large-Scale Structured Data

## Abstract

Last decade has seen the emergence of numerous methods for learning on graphs, particularly Graph Neural Networks (GNNs). These methods, however, are often not directly applicable to more complex structures like bipartite graphs (equivalent to hypergraphs), which represent interactions among two entity types (e.g., a user liking a movie). This paper proposes Convolutional Signal Propagation (CSP), a non-parametric simple and scalable method that natively operates on bipartite graphs (hypergraphs) and can be implemented with just a few lines of code. After defining CSP, we demonstrate its relationship with well-established methods like label propagation, Naive Bayes, and Hypergraph Convolutional Networks. We evaluate CSP against several reference methods on real-world datasets from multiple domains, focusing on retrieval and classification tasks. Our results show that CSP offers competitive performance while maintaining low computational complexity, making it an ideal first choice as a baseline for hypergraph node classification and retrieval. Moreover, despite operating on hypergraphs, CSP achieves good results in tasks typically not associated with hypergraphs, such as natural language processing.

## 1 Introduction

In the modern world, an overwhelming amount of data has an internal structure, oftentimes forming complex networks that can be represented as graphs. Efficiently mining information from this data is crucial for a wide range of applications spanning various domains such as social networks, biology, physics or cybersecurity. Graph Neural Networks (GNNs) have emerged as the dominant tool for handling such data due to their ability to leverage the graph structure for predictive and analytical tasks. However, despite their success, GNNs come with notable challenges, including high computational complexity during training, numerous hyperparameters that require fine-tuning, lack of straightforward interpretability, and the necessity of dedicated computational infrastructure such as GPUs. Given these limitations, baseline algorithms play a vital role as complementary tools to GNNs. These baselines, often much less complex, provide an efficient way of generating preliminary results. In many cases, these simpler methods are even sufficiently effective to be used as-is for the problem at hand.

In this work, we present Convolutional Signal Propagation (CSP)[1], a simple, scalable algorithm that may serve as such a baseline. We describe the algorithm in Section 4 and provide an overview of how it relates to established methods. While CSP is a general algorithm for propagating any signals and is introduced as such, we are mostly interested in its application to classification and retrieval (i.e., a setting similar to label propagation (Zhu & Ghahramani, 2003)) and provide experimental evaluation in this setting in Section 5.

## 2 Problem Statement

Assume that we have structured data with relationships that can be translated into a hypergraph or a bipartite graph. These structures can represent various scenarios such as users

---

[1] See https://anonymous.4open.science/r/CSP-demo-ICLR-2025 for a demo of CSP

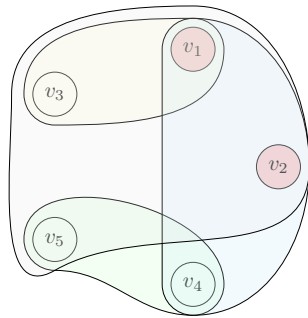
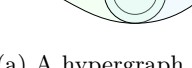

(a) A hypergraph.

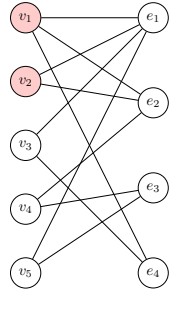

(b) A bipartite graph.

$$\boldsymbol{H} = \begin{bmatrix} 1 & 1 & 0 & 1 \\ 1 & 1 & 0 & 0 \\ 1 & 0 & 0 & 1 \\ 0 & 1 & 1 & 0 \\ 1 & 0 & 1 & 0 \end{bmatrix}$$

(c) An incidence matrix of a hypergraph.

Figure 1: An example of three different represetations of a dataset with 5 entities and 4 different relationships between them.

rating movies, users accessing web domains, emails containing attachments, authors writing papers, papers being co-cited, and tokens being contained in texts. Some data might also come with constraints such as large volume or privacy restrictions, like those found in emails. These structures inherently provide a relationship between entities, enabling many applications to leverage these relationships, such as movie recommendations, mining malicious web content (like emails or domains), or text classification.

Our goal is to find a flexible method that can be applied to tasks involving structured data. This flexibility is sought in terms of performance, numerical complexity, and adaptability. We aim to develop a method that can efficiently handle and extract meaningful information from these complex relationships. Whether we are looking to extract interesting entities, which aligns with a retrieval scenario, or view the task as a simple classification problem, the method should be versatile enough to adapt to these needs.

We can formalize this problem using either a bipartite graph or a hypergraph. By representing the data in these graph structures, we can better understand and utilize the inherent relationships between entities. This formalization allows for the application of graph-based algorithms, which can improve the effectiveness of tasks like classification and retrieval. Figure 1 shows an example of such representations.

## 2.1 NOTATIONS AND DEFINITIONS

We consider a finite set of items of interest $V = \{v_1, \ldots, v_n\}$, referred to as *nodes*. A family of subsets of $V$ denoted by $E = \{e_1, \ldots, e_m\} \subseteq 2^V$ is referred to as *hyperedges*. The nodes and hyperedges togerher form a *hypergraph* $\mathcal{H} = (V, E)$. The structure of the hypergraph can also be described by an *incidence matrix* $\boldsymbol{H} \in \{0,1\}^{n \times m}$, where $\boldsymbol{H}_{i,j} = 1$ if $v_i \in e_j$, and 0 otherwise. Every hypergraph can alternatively be described by a bipartite *incidence graph*, also called the Levi graph (Levi, 1942). This bipartite graph $\mathcal{G}_{\text{bip}} = (V \cup E, E_{\text{bip}})$ has as its two partitions the nodes and hyperedges of $\mathcal{H}$, and its edges represent a node in $V$ belonging to an edge in $E$, formally $E_{\text{bip}} = \{(v_i, e_j) \in V \times E | v_i \in e_j\}$.

The degree $\mathrm{d}(v)$ of node $v$ is defined as the number of edges that contain the node. Similarly, the degree $\delta(e)$ of the edge $e$ is defined as the number of nodes it contains. We also establish a diagonal node-degree matrix $\boldsymbol{D}_v \in \mathbb{N}^{n \times n}$ with $(\boldsymbol{D}_v)_{i,i} = \mathrm{d}(v_i)$ and $(\boldsymbol{D}_v)_{i,j} = 0$ for $i \neq j$. Analogously, the hyperedge-degree matrix is a diagonal matrix $\boldsymbol{D}_e \in \mathbb{N}^{m \times m}$ with $(\boldsymbol{D}_e)_{i,i} = \delta(e_i)$ and $(\boldsymbol{D}_e)_{i,j} = 0$ for $i \neq j$.

We consider for each node in the hypergraph some kind of signal that is to be propagated through the hyperedges. Let the signal be a $d$-dimensional vector $\boldsymbol{x}_i$ for each node, giving for the whole hypergraph a matrix $\boldsymbol{X} \in \mathbb{R}^{n \times d}$. In the following parts of this paper, we will explore several ways of defining such a signal, with an overview provided in Section 4.2.

Within this work, we are interested in two transductive tasks on hypergraphs: classification on $V$ and retrieval of positive nodes from $V$. Both tasks assume a training set of nodes $V_{\text{train}} \subset V$ where the labels of nodes are known. In case of classification, the goal is to predict the label for all nodes in $V$. The retrieval task aims to sort the nodes in the testing set $V \setminus V_{\text{train}}$ such that the number of positive nodes in top $K$ positions is maximized.

## 3 Related work

Mining information from structured (graph-like) data is one of the central problems in machine learning. The most straightforward way to handle this is to translate the structure into features and apply traditional machine learning techniques, such as logistic regression, random forests, and naive Bayes, to these features. Naive Bayes (Ng & Jordan, 2001), in particular, provides a bridge to a second large family of learning methods on graphs: Bayesian methods, where the graph forms a structure for modelling random variables. A critical problem associated with Bayesian methods is inference, which is often intractable.

The translation of structured data into features is also a non-trivial problem. While some methods can handle sparse, high-dimensional feature vectors, the majority cannot. Several methods are suited for finding low-dimensional representations of structured data. For example, non-negative matrix factorization (Lee & Seung, 2000) (NMF) decomposes a large sparse matrix into the product of two low-dimensional matrices. Other methods, such as node2vec (Grover & Leskovec, 2016), spectral positional encodings (Dwivedi et al., 2023) and distance encodings (Li et al., 2020; Beaini et al., 2021) offer node representations for graphs, however, they cannot be directly applied to hypergraphs.

There are many papers on learning algorithms for graphs, such as GraphSAGE (Hamilton et al., 2017), graph convolutional networks (Kipf & Welling, 2017), and graph attention networks (Veličković et al., 2018). Nevertheless, their application to hypergraphs is not straightforward. The origins of learning transductive tasks stretch back to the seminal work by Zhou et al. (2006). More recently, Hypergraph neural networks (Feng et al., 2019), Dynamic HGNNs (Jiang et al., 2019), and HyperGCN (Yadati et al., 2019) build upon the convolutional learning schema introduced in Kipf & Welling (2017) while works such as Bai et al. (2021) aim to bring both convolutional as well as attention to the context of hypergraphs. The proposed method can also be viewed as an extension of label propagation (Zhu & Ghahramani, 2003; Huang et al., 2020) or feature propagation (Rossi et al., 2022). While there do exists algorithm for label propagation in hypergraphs (Henne, 2015; Lee et al., 2024), the proposed method aims to be comparatively simpler to understand, implement and calculate.

## 4 Convolutional Signal Propagation

We present Convolutional Signal Propagation (CSP), a method for signal propagation on hypergraphs. In the following subsections, CSP is first introduced in the general setting, followed by a comparison to established approaches and a discussion of possible variants inspired by them. Finally, applications of CSP to different kinds of signals in hypergraph tasks are discussed.

### 4.1 Method overview

The proposed algorithm propagates a node signal $\boldsymbol{X}$ (see Section 4.2 for a discussion of possible signal types) through the hypergraph $\mathcal{H}$. The basic version of CSP consists in a simple averaging of $\boldsymbol{X}$ across the hyperedges and nodes of the graph. This averaging can be repeated to obtain smoother final representations, resulting in a multi-step CSP process generating a sequence of representations $\boldsymbol{X}^{(l)}$, where $\boldsymbol{X}^{(0)} = \boldsymbol{X}$. Appendix A.1 gives an example of how CSP is applied to a simple dataset.

In each step, the representation $\boldsymbol{X}^{(l)}$ of the nodes is first propagated to the hyperedges to obtain their representations

$$\boldsymbol{r}_j^{(l)} = \frac{1}{\delta\left(e_j\right)} \sum_{\substack{i \\ v_i \in e_j}} \boldsymbol{x}_i^{(l)} \tag{1}$$

that is the average of the representation of the individual nodes contained in the hyperedge. In the second step, this hyperedge representation is propagated again into nodes:

$$\boldsymbol{x}_k^{(l+1)} = \frac{1}{\mathrm{d}\left(v_k\right)} \sum_{\substack{j \\ v_k \in e_j}} \boldsymbol{r}_j^{(l)}. \tag{2}$$

The steps 1 and 2 constitute the proposed Convolutional Signal Propagation algorithm, which can be summarily written as

$$\boldsymbol{x}_k^{(l+1)} = \frac{1}{\mathrm{d}\left(v_k\right)} \sum_{\substack{j \\ v_k \in e_j}} \frac{1}{\delta\left(e_j\right)} \sum_{\substack{i \\ v_i \in e_j}} \boldsymbol{x}_i^{(l)}. \tag{3}$$

Using notation established in Section 2.1, Equation 3 can be rewritten into the matrix form

$$\boldsymbol{X}^{(l+1)} = \boldsymbol{D}_v^{-1} \boldsymbol{H} \boldsymbol{D}_e^{-1} \boldsymbol{H}^T \boldsymbol{X}^{(l)}. \tag{4}$$

Equation 4 describes a basic variant of the proposed algorithm. In Section 4.6, various modifications of CSP are discussed. While Equation 4 shows an efficient way of mathematically expressing the CSP algorithm, the algorithm itself is also efficient when it comes to its implementation and computational complexity. See Appendix A.2 for an overview of ways of implementing CSP. The asymptotic computational complexity of CSP may be observed from Equation 3 as

$$\mathcal{O}\left(d\left(\Sigma_V + \Sigma_E\right)\right) \tag{5}$$

where $d$ is the signal dimensionality, $\Sigma_V$ is the sum of node degrees and $\Sigma_E$ is the sum of hyperedge degrees. Of note is also the fact that Equation 4 preserves the sparsity of $\boldsymbol{H}$.

### 4.2 Application of CSP to different signals in hypergraphs

The construction of Convolutional Signal Propagation in Section 4.1 was a general one, assuming a signal matrix $\boldsymbol{X} \in \mathbb{R}^{n \times d}$. In practice, one can use CSP to propagate various kinds of "signals" in the hypergraph. Namely, the matrix $\boldsymbol{X}$ may represent actual node features as provided in the underlying graph dataset. This setting leads to a method similar to feature propagation (Rossi et al., 2022) or hypergraph convolution (Feng et al., 2019). Such an approach is elaborated further in Section 4.3. Alternatively, an approach based on label propagation by Zhu & Ghahramani (2003) may be obtained by taking as $\boldsymbol{X}$ a version of the label matrix $\boldsymbol{Y}$ masked by the training set, a setting described in Section 4.4 and 4.5 and evaluated in Section 5.

### 4.3 Comparison with Hypergraph Convolution

A single layer of the Hyper-Conv hypergraph neural network by Bai et al. (2021) is defined as

$$\boldsymbol{X}^{(l+1)} = \sigma(\boldsymbol{D}_v^{-1} \boldsymbol{H} \boldsymbol{W} \boldsymbol{D}_e^{-1} \boldsymbol{H}^T \boldsymbol{X}^{(l)} \boldsymbol{\Theta}), \tag{6}$$

where $\boldsymbol{W}$ and $\boldsymbol{\Theta}$ are weight matrices that need to be optimized.

Comparing Equations 4 and 6, it can be seen that CSP is a simplified special case of Hyper-Conv with the matrices $\boldsymbol{W}$ and $\boldsymbol{\Theta}$ realized as non-learnable identity matrices. As the proposed method runs only the "forward pass" of Hyper-Conv, we do not use the non-linearity $\sigma$ in the basic variant of CSP.

## 4.4 Comparison with Label Propagation

The label propagation algorithm as introduced in Zhu & Ghahramani (2003) is expressed for an ordinary graph (with edges connecting exactly 2 nodes) as

$$\boldsymbol{Y}^{(l+1)} = \alpha \boldsymbol{D}_v^{-1} \boldsymbol{A} \boldsymbol{Y}^{(l)} + (1-\alpha) \boldsymbol{Y}^{(l)}, \tag{7}$$

where $\boldsymbol{D}_v$ denotes the diagonal matrix of degrees of a graph and $\boldsymbol{A}$ stands for its adjacency matrix.

To compare Label propagation with CSP, let us first express the value of $\boldsymbol{H}\boldsymbol{D}_e^{-1}\boldsymbol{H}^T$ as

$$\left(\boldsymbol{H}\boldsymbol{D}_e^{-1}\boldsymbol{H}^T\right)_{i,j} = \sum_k \frac{1}{\delta(e_k)} \boldsymbol{H}_{i,k}\boldsymbol{H}_{j,k}, \tag{8}$$

which represents for each pair of nodes the number of hyperedges connecting them, normalized by their degrees. Specifically, for an ordinary graph, this becomes

$$\boldsymbol{H}\boldsymbol{D}_e^{-1}\boldsymbol{H}^T = \frac{1}{2}\left(\boldsymbol{A} + \boldsymbol{D}_v\right). \tag{9}$$

With this simplification for ordinary graphs, Equation 4 becomes

$$\boldsymbol{X}^{(l+1)} = \frac{1}{2}\boldsymbol{D}_v^{-1}\boldsymbol{A}\boldsymbol{X}^{(l)} + \frac{1}{2}\boldsymbol{X}^{(l)}. \tag{10}$$

which is equivalent to Equation 7 with $\boldsymbol{X} = \boldsymbol{Y}$ (or a masked version thereof) and $\alpha = \frac{1}{2}$. CSP is in this instance therefore a generalization of label propagation with this particular value of $\alpha$ to hypergraphs (for generalization with arbitrary values of $\alpha$, see Section 4.6). There is, however, another compelling reason to use CSP over Label propagation as presented in Equation 7. The matrix multiplication $\boldsymbol{H}\boldsymbol{D}_e^{-1}\boldsymbol{H}^T$ does not preserve the sparsity of $\boldsymbol{H}$, which is typical for large datasets. Therefore the proposed implementation can be significantly more efficient than Equation 7, despite them being mathematically equivalent.

## 4.5 Comparison with the Naive Bayes classifier

The Naive Bayes classifier is a well-known classification method that calculates the posterior probability $p(y|\boldsymbol{\xi})$ of a label $y$ given a feature vector $\boldsymbol{\xi}$. Using Bayes' rule, this is expressed as $p(y|\boldsymbol{\xi}) = p(\boldsymbol{\xi}|y)p(y)/p(\boldsymbol{\xi})$ with the "naive" assumption that the conditional probability $p(\boldsymbol{\xi}|y)$ can be factorized as $p(\boldsymbol{\xi}|y) = \prod_i p(\xi_i|y)$. Here, $p(\xi_i|y)$ is estimated from the training set for all feature-label pairs.

In the case of binomial Naive Bayes, the maximum likelihood estimation of the model parameter $p(\xi_i|y=1)$ is given by the ratio of the number of positive examples containing feature $\xi_i$ to the total number of examples in the training set that include $\xi_i$. When considering nodes as examples and features as hyperedges, the estimated binomial parameter of a given feature corresponds to the hyperedge score defined in Equation 1. While Naive Bayes inference is based on probabilistic reasoning, where predictions are the product of the model parameters assuming feature independence, CSP employs a filtering (convolutional) approach. Applying Bayes' rule translates Naive Bayes into a posterior prediction reflecting the prior distribution, which is not captured by CSP. Therefore, Naive Bayes is expected to perform better in classification tasks. On the other hand, accounting for priors becomes a disadvantage in a retrieval setup, as target examples typically have low prior probabilities.

## 4.6 Convolutional Signal Propagation Extensions

The comparison with the methods presented in Sections 4.3, 4.4 and 4.5 naturally suggests several alternative variants and generalizations of the basic CSP scheme. All of them can be implemented in a straightforward way by modifying Equation 4, without requiring full matrix multiplication.

### 4.6.1 ALTERNATIVE NORMALIZATIONS OF THE ADJACENCY MATRIX

In graph neural networks, the adjacency matrix $\boldsymbol{A}$ is normalized by multiplying it with the inverse of the node degree matrix $\boldsymbol{D}_v$. While the DeepWalk algorithm (Perozzi et al., 2014) corresponds to the row-wise normalization $\boldsymbol{D}_v^{-1}\boldsymbol{A}$, newer methods also consider the column-wise normalization $\boldsymbol{A}\boldsymbol{D}_v^{-1}$ and most predominantly the symmetric normalization $\boldsymbol{D}_v^{-1/2}\boldsymbol{A}\boldsymbol{D}_v^{-1/2}$ introduced in Kipf & Welling (2017). Because the matrix $\boldsymbol{H}\boldsymbol{D}_e^{-1}\boldsymbol{H}^T$ plays in CSP a role similar to the adjacency matrix $\boldsymbol{A}$ in GCN (see Equation 8), we can also consider the alternative column-wise normalized version of CSP:

$$\boldsymbol{X}^{(l+1)} = \boldsymbol{H}\boldsymbol{D}_e^{-1}\boldsymbol{H}^T\boldsymbol{D}_v^{-1}\boldsymbol{X}^{(l)} \tag{11}$$

and the symmetrically normalized version

$$\boldsymbol{X}^{(l+1)} = \boldsymbol{D}_v^{-1/2}\boldsymbol{H}\boldsymbol{D}_e^{-1}\boldsymbol{H}^T\boldsymbol{D}_v^{-1/2}\boldsymbol{X}^{(l)}. \tag{12}$$

### 4.6.2 GENERALIZATION OF LABEL PROPAGATION WITH GENERAL VALUES OF $\alpha$

Section 4.4 shows that CSP is generalization of label propagation with $\alpha = \frac{1}{2}$ to hypergraphs. This parameter $\alpha$ in label propagation controls how much the score of a node is influenced by its neighbors. To provide such a configurability for CSP, a similar parameter $\alpha'$ may be introduced:

$$\boldsymbol{X}^{(l+1)} = \alpha'\boldsymbol{D}_v^{-1}\boldsymbol{H}\boldsymbol{D}_e^{-1}\boldsymbol{H}^T\boldsymbol{X}^{(l)} + (1 - \alpha')\boldsymbol{X}^{(l)}. \tag{13}$$

This version is a full generalization of label propagation as described in Equation 7 to hypergraphs. Note that the parameter $\alpha'$ has different semantics and boundary values from $\alpha$ in label propagation.

### 4.6.3 CSP IN INDUCTIVE SETTINGS

Within the definition of CSP in Equation 3, we assume that the algorithm operates on a fixed hypergraph $\mathcal{H}$, which corresponds to a transductive setup. This means the algorithm requires the entire hypergraph to be processed at once.

By decomposing CSP into its stages, we can adapt it for the inductive scenario. The score calculated according to Equation 1 can be treated as a trained model, similar to the way a Naive Bayes model is trained (see Section 4.5). The second stage (Equation 2) can then be applied to the testing data, where the node of interest does not need to have been part of the training process, thus allowing for induction. Another advantage is that the second stage of CSP (Equation 2) is performed independently for each node.

The inductive extension also offers several options. First, while Equation 2 naturally handles nodes that did not appear in the original hypergraph $\mathcal{H}$, incorporating new hyperedges is less straightforward. These new hyperedges can either be ignored, or their scores can be additionally evaluated using Equation 1 and incorporated alongside the trained model. When considering multiple layers, another degree of flexibility emerges, as model training and prediction can be applied at arbitrary layer.

## 5 APPLICATIONS AND EXPERIMENTAL EVALUATION

The goal of our experiments is twofold. We first aim to demonstrate the versatility of CSP by applying it to problems from multiple domains, and second, we would like to compare the performance and execution time of CSP with several well established baseline methods as well as with a simple Hypergraph Neural Network (HGCN). Our aim is to validate the comparable performance of the proposed method while highlighting its low execution time. While we discuss the various extensions of the proposed method in Section 4.6, their comprehensive evaluation is left for future work.

### 5.1 DATASETS

CSP operates on hypergraphs, and we exemplify several problems from various domains that can be represented as hypergraphs suitable for CSP application. The considered datasets

Table 1: Overview of datasets with their basic characteristics. $\Sigma_E$ is the sum of hyperedge degrees or equivalently the number of non-zero elements in the incidence matrix $\boldsymbol{H}$. Isolated nodes are nodes that are not connected by any hyperedge. Percentage of isolated nodes is their fraction related to the overall number of nodes.

| Dataset | CiteSeer | Cora-CA | Cora-CC | DBLP | PubMed | Corona | movie-RA | movie-TA |
|---|---|---|---|---|---|---|---|---|
| **Node** | paper | paper | paper | paper | paper | text | movie | movie |
| **Node label** | topic | topic | topic | topic | topic | sentiment | category | category |
| **Hyperedge** | citation | author | citation | author | citation | token | user | tag |
| **Nodes** | 3312 | 2708 | 2708 | 41302 | 19717 | 44955 | 62423 | 62423 |
| **Isolated nodes** | 1854 | 320 | 1274 | 0 | 15877 | 0 | 3376 | 17172 |
| **Hyperedges** | 1079 | 1072 | 1579 | 22363 | 7963 | 998 | 162541 | 14592 |
| **$\Sigma_E$** | 3453 | 4585 | 4786 | 99561 | 34629 | 3455918 | 25000095 | 1093360 |
| **Average** $d(v)$ | 2.37 | 1.92 | 3.34 | 2.41 | 9.02 | 76.88 | 423.39 | 24.16 |
| **Average** $\delta(e)$ | 3.20 | 4.28 | 3.03 | 4.45 | 4.35 | 3463 | 153.8 | 74.9 |
| **Classes** | 6 | 7 | 7 | 6 | 3 | 5 | 20 | 20 |

are summarized in Table 1, including basic statistics and details on how they are translated into hypergraphs. These datasets may be grouped into three families.

The first family consists of citation networks such as PubMed, Cora, and DBLP. Their hypergraph variants, as introduced in Chien et al. (2021), are based on hyperedges defined by sets of papers either sharing the same author or being cited in the same paper. Each publication is labeled based on its topic.

The second family includes datasets represented by one-hot features. Specifically, we consider the Coronavirus tweets dataset (Corona) (Miglani, 2020), which contains Twitter posts about COVID-19. The tweets are tokenized, with each tweet being represented as a node labeled by its sentiment. Hyperedges are formed by collections of tweets sharing the same token. We apply the Sentencepiece tokenization algorithm (Kudo & Richardson, 2018) on the entire corpus with a target of 1000 tokens[2]. Note that we can control the graph size (i.e., the number of hyperedges) through the tokenization parameter.

Finally, we consider the MovieLens 25M dataset (Harper & Konstan, 2015), which contains 25 million user ratings for movies and one million tagged movies. Hyperedges are formed by collections of movies either rated by the same user or sharing the same tag. The nodes represent individual movies, labeled by their genres (with multiple labels allowed per node).

Due to the interpolative nature of CSP, each multiclass dataset is transformed into a series of binary datasets, where one class is treated as the positive class, and all other classes are treated as negative. The results are averaged over all the resulting binary datasets. Only structural information available in the hypergraph is considered; no additional information such as node or hyperedge features are included.

### 5.2 Tasks

We address two primary tasks in our experiments: transductive node classification and retrieval.

#### 5.2.1 Classification Task

In the classification setting, the aim of the model is to predict binary labels on the testing set. We use leave-one-out cross-validation with 10 folds, where nodes are randomly assigned to folds. One fold is hidden for testing, and the method is trained on the remaining nine folds. For each dataset, we generate test predictions for each node (when it is in the testing set). For each class, we calculate the ROC-AUC and average these scores. This average is reported as the classification score for each method on the given dataset.

---

[2]The number of hyperedges in Table 1 differs due to reserved tokens not used in the corpus.

### 5.2.2 Retrieval Task

In the retrieval setting, the aim of the model is to rank the nodes in the test set to maximize precision in the top positions. In this case, one fold is used for training and the other 9 are used as the testing dataset. The training set consists of all nodes in the graph, with the positive nodes in the training fold labeled as positive and all other nodes labeled as unknown. For models that require negative training examples, we randomly sample a set of the same size as the testing set and consider these labels as negative. Although this introduces some label noise, we assume that the negative class is dominant, making the noise acceptable. The model then ranks the nodes from the testing folds, and we evaluate precision at the top 100 positions (P@100). This evaluation is performed for each fold and class, and the average P@100 over folds and classes is reported as the retrieval score for each method on the given dataset.

### 5.3 Evaluated Methods

The goal of this work is to show the comparative performance of CSP and its computational efficiency. To this end, the basic variant of CSP is compared to the following methods. In future, we would like to also compare the proposed modifications of CSP mentioned in Section 4.6 and multiple choices of feature representation for reference methods on top of NMF.

- **The proposed CSP method**: Evaluated with 1, 2, and 3 layers, where we consider binary training labels as $X^0$. After application of a given number of CSP layers (see Equation 4), i.e., the yielded (score) vector $X^l, l \in \{1, 2, 3\}$ is used for both retrieval (top-100 scored test nodes) and for binary classification (with a given threshold on score).

- **Multinomial Naive Bayes**: Operates on one-hot feature vectors derived from hyperedges.

- **Random Forest, Logistic Regression, and HGCN**: These methods operate on feature vectors obtained from non-negative matrix factorization (NMF) of the incidence matrix (Lee & Seung, 2000)[3] $H$, with 10 iterations and a dimension of 60. To configure the methods themselves, Random Forest, Logistic Regression, and Naive Bayes are used with their default settings. For HGCN, a single layer implementing Equation (6) was used, with an output layer of dimension 2 and sigmoid non-linearity. We use logistic loss and train all datasets for 15,000 epochs using the Adam optimizer with default settings.

- **Random Baseline**: Included for comparison.

The results of these methods are evaluated and compared based on their performance on the classification and retrieval tasks across the datasets.

### 5.4 Classification Results

Table 2 lists the ROC-AUC for all methods on all datasets. Due to the numerical intensity of HGCN, only 5 folds were evaluated, and for the Movies dataset, only 4 out of 20 classes were considered. Prediction on isolated nodes was nearly random as only structural information was used, likely contributing to the relatively weak performance of all methods on the PubMed dataset. Since CSP handles only binary labels, reference methods were translated to a one-vs-other scenario, even though they can handle multi-class classification directly. Feature extraction using Non-negative Matrix Factorization (NMF) was not fine-tuned for each dataset, potentially impacting the performance of NMF-based baselines. Naive Bayes emerged as the strongest baseline, as it does not require any feature preprocessing and works directly with the one-hot encoded incidence matrix. CSP was evaluated in three variants

---

[3]As an alternative to the NMF, we evaluated also representation generated by Laplacian positional encoding (Dwivedi et al., 2023). As the results were worse compared to NMF, we decided to not include them in the results.

Table 2: The ROC-AUC for the classification task, averaged over all classes and all folds. The best method for each dataset is denoted by bold text, with methods within 0.05 underlined.

| Method | CiteSeer | Cora-CA | Cora-CC | DBLP | PubMed | Corona | movie-RA | movie-TA |
|---|---|---|---|---|---|---|---|---|
| **CSP 1-layer** | 0.646 | 0.882 | 0.716 | 0.968 | 0.537 | **0.704** | 0.789 | 0.717 |
| **CSP 2-layer** | 0.630 | 0.872 | 0.686 | 0.972 | 0.518 | 0.618 | 0.700 | 0.697 |
| **CSP 3-layer** | 0.613 | 0.862 | 0.655 | 0.972 | 0.516 | 0.580 | 0.640 | 0.673 |
| **Naive Bayes** | **0.686** | **0.913** | 0.775 | **0.974** | **0.633** | **0.704** | 0.753 | 0.557 |
| **HGCN-NMF** | 0.659 | 0.832 | **0.786** | 0.775 | 0.624 | 0.622 | 0.794 | **0.724** |
| **LR-NMF** | 0.604 | 0.794 | 0.703 | 0.705 | 0.556 | 0.647 | 0.754 | 0.675 |
| **RF-NMF** | 0.667 | 0.897 | 0.772 | 0.905 | 0.623 | 0.617 | **0.797** | 0.691 |
| **Random** | 0.499 | 0.505 | 0.489 | 0.501 | 0.502 | 0.503 | 0.499 | 0.487 |

Table 3: The P@100 for the retrieval task, averaged over all classes and all folds. The best method for each dataset is denoted by bold text, with methods within 0.05 underlined.

| Method | CiteSeer | Cora-CA | Cora-CC | DBLP | PubMed | Corona | movie-RA | movie-TA |
|---|---|---|---|---|---|---|---|---|
| **CSP 1-layer** | 0.494 | 0.703 | 0.530 | 0.869 | 0.798 | **0.530** | 0.334 | 0.156 |
| **CSP 2-layer** | 0.558 | 0.718 | 0.681 | 0.865 | 0.826 | 0.440 | 0.336 | 0.186 |
| **CSP 3-layer** | **0.568** | **0.721** | **0.707** | 0.869 | 0.850 | 0.332 | 0.238 | 0.186 |
| **Naive Bayes** | 0.471 | 0.686 | 0.491 | **0.951** | 0.860 | 0.446 | 0.216 | 0.153 |
| **HGCN-NMF** | 0.482 | 0.671 | 0.607 | 0.794 | **0.871** | 0.392 | 0.257 | 0.148 |
| **LR-NMF** | 0.329 | 0.603 | 0.372 | 0.602 | 0.735 | 0.397 | **0.580** | **0.356** |
| **RF-NMF** | 0.303 | 0.474 | 0.482 | 0.843 | 0.794 | 0.381 | 0.470 | 0.131 |
| **Random** | 0.153 | 0.132 | 0.129 | 0.155 | 0.308 | 0.180 | 0.040 | 0.055 |

based on the number of layers, with the best choice varying by dataset. On the largest datasets (Corona and Movies), the best variant of CSP achieved performance comparable to the strongest competing baseline. Overall, CSP demonstrated comparable results with reference baselines. In larger datasets, where parameter tuning of baselines is more challenging, CSP proved to be one of the best-performing methods. Overall, CSP with fewer layers fared comparatively better than a version with multiple layers. We attribute this at first glance counter-intuitive result to the fact that the training set is fairly dense in the graph, which ensures sufficient information for all nodes even with fewer layers, while at the same time multiple layers may contribute to oversmoothing of the signal. These results confirm the suitability of CSP as a first-choice baseline method for classification tasks.

## 5.5 Retrieval Results

Table 3 lists the P@100 for all methods on all datasets. The evaluation of HGCN and the Movies dataset in the retrieval task is restricted similarly to the classification task. Isolated nodes no longer cause a performance drop as long as there are sufficient number of non-isolated nodes in each class. Naive Bayes' performance is not as superior in this scenario as in classification task since the training set contains only positive nodes, preventing it from leveraging prior distribution knowledge about the target class. CSP, which does not use prior knowledge about the target class distribution, works very well on small datasets with lower average degree of the nodes and edges. In case of datasets with higher average degree of nodes (Movies), CSP does not extract the structural information as well as NMF and therefore the methods utilizing the features from NMF (mainly logistic regression) work much better except HGCN, which suffers from over-smoothing. In summary, CSP achieves superior performance for 4 of 8 datasets on retrieval task and is significantly worse on only the Movies dataset, showing its suitability as a baseline in the retrieval setting.

## 5.6 Complexity Evaluation

The wall-clock execution time for individual methods is presented in Table 4. We evaluated the methods using standard implementations that would be widely used by practitioners. In particular, we used the Scikit-Learn (Pedregosa et al., 2011) implementation with de-

Table 4: The execution time of a single retrieval task in microseconds, averaged over all classes and all folds. The non-negative matrix factorization was excluded from the execution time.

| Method | CiteSeer | Cora-CA | Cora-CC | DBLP | PubMed | Corona | movie-RA | movie-TA |
|---|---|---|---|---|---|---|---|---|
| **CSP 1-layer** | **1.35** | **1.23** | **1.24** | **3.51** | **4.01** | **22.83** | 170.63 | **12.07** |
| **CSP 2-layer** | 2.41 | 2.25 | 2.24 | 7.46 | 7.35 | 47.39 | 349.67 | 25.88 |
| **CSP 3-layer** | 3.31 | 3.09 | 3.08 | 9.65 | 9.1 | 70.98 | 506.1 | 35.75 |
| **Naive Bayes** | 2.59 | 2.73 | 2.54 | 11.16 | 5.35 | 89.3 | 1 051.53 | 35.61 |
| **HGCN-NMF** | 23 714 | 23 690 | 23 709 | 29 123 | 23 879 | 112 314 | 620 717 | 70 778 |
| **LR-NMF** | 44.45 | 51.59 | 49.54 | 64.96 | 44.69 | 56 | **61.82** | 74.07 |
| **RF-NMF** | 140.76 | 143.85 | 134.52 | 1 148.83 | 323.6 | 1 608.58 | 697.85 | 716.68 |

fault settings for logistic regression, Naive Bayes and random forest. A Polars variant of Equation 4 (see Appendix A.2) was applied for the proposed CSP method. All these methods were executed on a GPU (Amazon EC2 G4 instance). The HGCN was executed using PyTorch-geometric (Fey & Lenssen, 2019). Comparing the execution times, HGCN is the most numerically complex method. Although methods to improve training efficiency are available, they were not considered in this work. Logistic regression and random forest exhibit relatively short execution times in Corona and Movies datasets, largely because the most challenging part—extraction of structural information—is handled by nonnegative matrix factorization, which is not included in the reported times. The proposed CSP excels particularly in graphs with a low average degree of nodes, similar to Naive Bayes and is roughly four orders of magnitude faster compared to the state-of-the-art HGCN. The measured execution time of CSP aligns with its expected asymptotic complexity (see Equation 5) and appears to be linear with the number of CSP layers, as anticipated.

In summary, HGCN is by far the most numerically intensive method and shows potential in some examples; however, there is still a significant amount of fine-tuning needed to achieve superior performance across datasets. NMF-based methods work exceptionally well on the Movies dataset for the retrieval task, though further tuning is required to properly extract structural information. Compared to Naive Bayes, CSP is the simpler method and is parameter-free. In some problems, CSP outperformed Naive Bayes and vice versa. Thus, both of these methods should be considered when establishing baselines for tasks on structural data.

## 6 CONCLUSION

This paper presents a signal propagation algorithm termed Convolutional Signal Propagation (CSP). We formally describe the CSP algorithm and demonstrate its simplicity and efficiency of implementation. This formal description allows us to show clear relationships between CSP and well-known algorithms such as Naive Bayes, label propagation, and hypergraph convolutional networks. These relationships suggest various algorithmic variants, which we left for detailed future exploration.

We discuss the application of CSP to different types of signals. Our primary focus is propagating binary labels, which is used for classification and retrieval tasks, positioning CSP as a hypergraph variant of traditional label propagation. Additionally, propagating node features instead of labels as signals leads to feature propagation. This dual functionality showcases the versatility of CSP in handling various tasks on hypergraphs.

The application of CSP to several real-world datasets from multiple domains demonstrates how these problems can be effectively expressed using hypergraphs. Evaluating CSP in these scenarios shows its competitive performance in both node classification and retrieval tasks, compared to a range of reference methods. Furthermore, we assess the computational complexity of these methods by examining their execution times, highlighting the simplicity and efficiency of CSP. This combination of competitive performance, low computational complexity, parameter-free nature, and flexibility in implementation makes CSP an ideal choice as a baseline for learning on structured data.

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

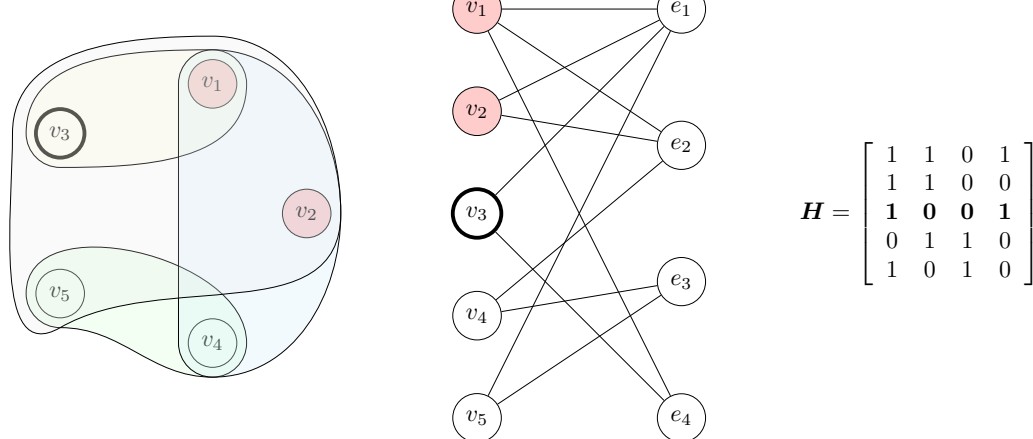

Figure 2: The node of interest $v_3$ highlighted in all three possible representations of the sample dataset.

Naganand Yadati, Madhav Nimishakavi, Prateek Yadav, Vikram Nitin, Anand Louis, and Partha Talukdar. HyperGCN: A New Method For Training Graph Convolutional Networks on Hypergraphs. In *Advances in Neural Information Processing Systems*, volume 32. Curran Associates, Inc., 2019. URL https://proceedings.neurips.cc/paper/2019/hash/1efa39bcaec6f3900149160693694536-Abstract.html.

Dengyong Zhou, Jiayuan Huang, and Bernhard Schölkopf. Learning with hypergraphs: Clustering, classification, and embedding. *Advances in neural information processing systems*, 19, 2006. URL https://proceedings.neurips.cc/paper/2006/hash/dff8e9c2ac33381546d96deea9922999-Abstract.html.

Xiaojin Zhu and Zoubin Ghahramani. Learning from Labeled and Unlabeled Data with Label Propagation. July 2003.

# A   Appendix

## A.1   An example of CSP computation

In this appendix, we give an example of how CSP is applied to the dataset from Figure 1. The dataset consists of 5 nodes (entities of interest) and 4 hyperedges (i.e., relationships between the entitites). While the description of CSP in Section 4.1 describes the algorithm as starting with some signal $\boldsymbol{X}^{(l)}$ in the nodes and propagates it through the hypergraph to obtain the updated signal $\boldsymbol{X}^{(l+1)}$, in this section, let us study the algorithm in reverse – i.e., how CSP calculates the updated score for a particular node.

Let us consider the node $v_3$, which is highlighted in Figure 2. In order to calculate its score $\boldsymbol{x}_3^{(l+1)}$, an obvious choice is to aggregate the scores of its incident edges $e_1$ and $e_4$. We consider the arithmetic mean as a form of such aggregation, giving us the relationship

$$\boldsymbol{x}_3^{(l+1)} = \frac{\boldsymbol{r}_1^{(l)} + \boldsymbol{r}_1^{(l)}}{2} = \frac{1}{\deg(v_3)} \sum_{\substack{j \\ v_3 \in e_j}} \boldsymbol{r}_j^{(l)} \tag{14}$$

where $\boldsymbol{r}_j^{(l)}$ is a score assigned to the edge $e_j$. This aggregation of edges scores into a node score is highlighted in Figure 3.

Naturally, the question of obtaining such edge scores arises. To this end, a similar approach may be applied, calculating the edge scores as simple arithmetic means of the previous

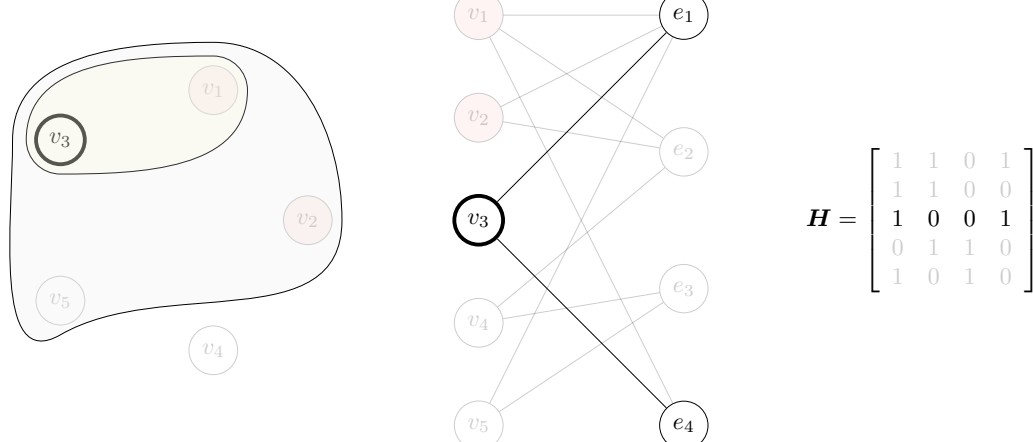

Figure 3: The node of interest $v_3$ and its incident edges. In CSP, the score of $v_3$ is obtained by averaging the scores of $e_1$ and $e_4$

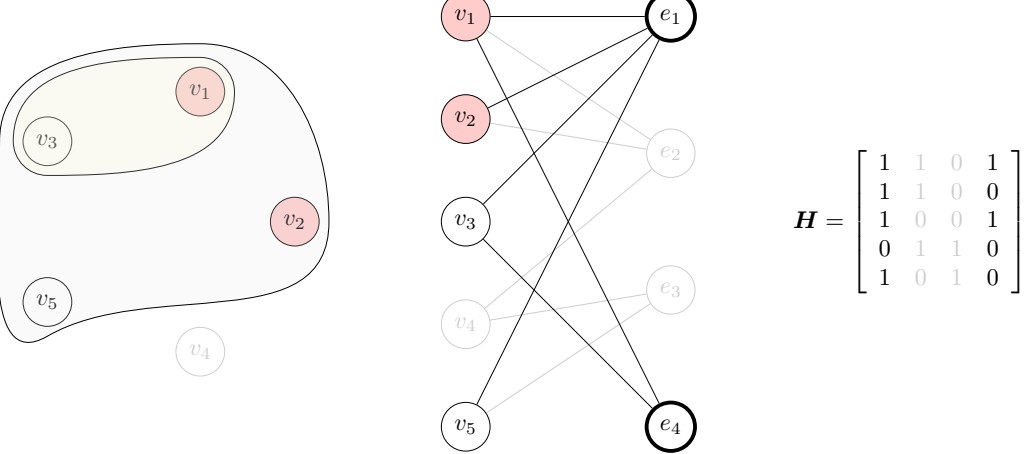

Figure 4: The edges $e_1$ and $e_4$ and their constituent nodes. In CSP, the score of $e_1$ is obtained by averaging the scores of $v_1$, $v_2$, $v_3$ and $v_5$, while the score of $e_4$ is obtained by averaging the scores of $v_1$ and $v_3$.

scores of nodes contained in each edge. In our particular dataset, this approach yields the following relationships

$$r_1^{(l)} = \frac{x_1^{(l)} + x_2^{(l)} + x_3^{(l)} + x_5^{(l)}}{4} = \frac{1}{\delta(e_1)} \sum_{\substack{i \\ v_i \in e_1}} x_i^{(l)} \tag{15}$$

$$r_4^{(l)} = \frac{x_1^{(l)} + x_3^{(l)}}{2} = \frac{1}{\delta(e_4)} \sum_{\substack{i \\ v_i \in e_4}} x_i^{(l)}. \tag{16}$$

This aggregation of node scores into edge scores is highlighted in Figure 4. Together with the previously described step of aggregating these edge scores into new node scores, the basic mechanism of CSP as outlined in Equation 3 is formed.

## A.2 On Efficient Implementation of Convolutional Signal Propagation

While Equation 4 shows an efficient way of mathematically expressing CSP, the algorithm itself is also efficient when it comes to its implementation and computational complexity. Algorithm 1 shows an implementation of CSP in a single SQL query, Algorithm 2 shows a simple implementation in Python using the Pandas (McKinney, 2010) library. Notably, the SQL implementation can also be applied using the Polars library (Vink et al., 2024), which was used for the experiments in Section 5.

These implementations essentially materialize Equations 1 and 2. The hypergraph $\mathcal{H}$ is represented as a table or DataFrame with columns *nodeId* and *edgeId*, where each row represents a given node belonging to a given hyperedge. The input table or DataFrame for CSP also contains a *nodeProperty* (signal) column, which is propagated through the method. The updated *nodeProperty* can then be used for either subsequent CSP layers or for the final prediction.

---

**Algorithm 1** An SQL implementation of a single CSP layer (3). Stage 1 and Stage 2 can be repeated multiple times before final aggregation when considering multiple layers.

---

```sql
WITH stage1 AS (
    SELECT nodeId, edgeId, AVG(nodeProperty) OVER (PARTITION BY edgeId) AS
        edgeProperty
    FROM table
),
stage2 AS (
    SELECT nodeId, edgeId, AVG(edgeProperty) OVER (PARTITION BY nodeId) AS
        nodeProperty
    FROM stage1
)
select nodeId, AVG(nodeProperty) as final_prediction from stage2
group by nodeId
```

---

**Algorithm 2** A Pandas implementation of CSP (3). The *CSP_layer* function may be applied to the DataFrame repeatedly before extracting the final prediction.

---

```python
import pandas as pd
def CSP_layer(df: pd.DataFrame) -> pd.DataFrame:
    df['edgeProperty'] = df.groupby('edgeId')['nodeProperty'].transform('mean')
    df['nodeProperty'] = df.groupby('nodeId')['edgeProperty'].transform('mean')
    return df

CSP_layer(df).groupby('nodeId').agg(final_prediction=('nodeProperty', 'mean'))
```

---

