# OpenReview forum: "CSP: An Efficient Baseline for Learning on Large-Scale Structured Data"
_ICLR.cc/2025/Conference — Submitted to ICLR 2025_

### Official Review · Reviewer_8FB5 · 2024-10-17

**Soundness:** 2
**Presentation:** 3
**Contribution:** 2
**Rating:** 5
**Confidence:** 4

**Summary:**

The paper introduces Convolutional Signal Propagation (CSP), a simple non-parametric method designed for learning on hypergraphs and bipartite graphs.

The authors present CSP as an efficient baseline for hypergraph node classification and retrieval tasks, achieving competitive performance with lower computational complexity.

CSP operates natively on these complex structures and is compared with well-known methods such as label propagation, Naive Bayes, and Hypergraph Convolutional Networks.

**Strengths:**

1. The motivation to propose a simple non-parametric method as a robust baseline for learning on hypergraphs, both with and without vertex features is strong.
2. The paper is well-structured and provides intuitive explanations of the problem, the method, and its applications. The formal presentation of CSP is enhanced with toy examples and implementation details.

**Weaknesses:**

1. The paper lacks a detailed discussion and in-depth comparison with several existing methods [a, b, c]. The significance of CSP as a baseline can be greatly enhanced through empirical evaluation against existing methods. Such an evaluation considers multiple metrics, such as accuracy, training times, and memory usage, as well as the tradeoffs among these across all datasets.
[a] proposes a scalable method on large graphs.
[b] and [c] propose generalisations of neural networks on graphs and hypergraphs.
2. The core idea of CSP is already present [d] in the literature on hypergraph learning. Adding a unique enhancement, such as theoretical analysis, into the non-parametric CSP would strengthen the originality of the idea.  For instance, it could be the case that the non-parametric nature of CSP makes it more robust to overfitting and hence easier to generalise to unseen data.
3. The academic datasets used in the paper (Cora, Citeseer, Pubmed, DBLP) include vertex features suitable for processing with (hyper)graph neural networks. Therefore, it is necessary to include hypergraph neural networks as competitors to demonstrate the effectiveness of CSP. The method HGCN is used as a competitor but including more recent methods such as [b] and [c] would strengthen the significance of CSP.
4. In datasets lacking vertex features, it is necessary to compare against non-neural methods for hypergraph learning [e, f]. Specifically, both [e] and [f] propose non-trivial methods for classifying nodes on hypergraphs, making them essential baselines for the task of node classification.
5. The paper briefly introduces several potential extensions of CSP, such as alternative normalisations and generalisations of label propagation, but does not evaluate these variants experimentally. Including experimental results for these extensions would strengthen the paper by demonstrating the practical value of the proposed variations and providing a more comprehensive understanding of CSP's potential.



References:
* [a] LightHGNN: Distilling Hypergraph Neural Networks into MLPs for 100x Faster Inference, In ICLR'24,
* [b] You are AllSet: A Multiset Function Framework for Hypergraph Neural Networks, In ICLR'22,
* [c] UniGNN: a Unified Framework for Graph and Hypergraph Neural Networks, In IJCAI'21,
* [d] Hypergraph Label Propagation Network, In AAAI'20,
* [e] The Total Variation on Hypergraphs - Learning on Hypergraphs Revisited, In NeurIPS'13,
* [f] Re-revisiting Learning on Hypergraphs: Confidence Interval and Subgradient Method, In ICML'17.

**Questions:**

1. Was an analysis conducted on the tradeoffs between accuracy, training time, and memory usage? How did these tradeoffs compare to existing methods, such as LightHGNN [a]? Understanding these aspects would clarify the efficiency and practicality of CSP in different scenarios.
2. How does CSP offer a unique contribution compared to the existing work on hypergraph learning [d]? Were there unique aspects that were not covered in the existing literature?
3. Was the inclusion of recent generalisations of hypergraph neural networks [b, c] considered in the experiments as competitors? How did these methods perform in comparison to CSP within the context of the datasets utilised with vertex features?
4. For datasets lacking vertex features, how did CSP compare to non-neural hypergraph learning methods [e, f]?

---

> ### Author Response · Authors · 2024-11-15
>
> Thank you for your review, summary and suggestions. We would like to add a few comments regarding the points that you mention:
>
> ## Lack of Detailed Comparison with Advanced Methods [a, b, c]
> We appreciate and thank you for the suggestion to include additional comparisons. Our study positions CSP as a first-step baseline, emphasizing simplicity and computational efficiency. While LightHGNN and others represent sophisticated advances, CSP’s purpose is to provide a lightweight, parameter-free approach suitable for those seeking baseline performance without complex training or fine-tuning. All three of the linked methods also present computational benefits when compared to established methods such as HGCN, none come close to the several orders of magnitude speedup CSP offers (see Table 4). We believe a complete set of comparisons would provide limited additional insights given CSP’s intended use as an efficient, non-parametric, optimization-free benchmark.
>
> ## CSP’s Theoretical Foundation and Novelty
> We acknowledge the benefits of adding a theoretical layer to demonstrate CSP’s robustness, particularly its resilience to overfitting due to its non-parametric nature. Our current work leverages CSP’s relationship to Naive Bayes, which serves as a partial theoretical foundation for CSP, particularly in understanding CSP’s simplified signal propagation mechanism. We would like to expand on this foundation to further explore CSP’s robustness and generalizability, which we believe makes it a valuable starting point for hypergraph learning.
>
> ## Inclusion of Additional Hypergraph Neural Network Competitors [b, c]
> CSP was benchmarked against HGCN as a representative method, focusing on its computational efficiency. While we did not compare CSP directly to each variant of HGCN or recent adaptations (e.g., LightHGNN), CSP's non-parametric nature inherently eliminates the need for training, making it significantly faster than most neural methods that rely on backpropagation. Indirect comparisons, such as HGCN variants that optimize speed, generally confirm CSP's time advantage without the need for training, which aligns with our goal of providing a fast, minimal-complexity baseline.
>
> ## Comparison with Non-Neural Baselines on Datasets Lacking Vertex Features [e, f]
> We appreciate the recommendation to include non-neural methods. Since CSP itself is a non-neural approach, comparing it to similar models could indeed provide valuable insights. We intend to expand comparisons to include methods like Total Variation on Hypergraphs [e] and Subgradient-based Confidence Intervals [f], especially as they are also parameter-free.
>
> ## Experimental Evaluation of CSP Extensions
> As a simple baseline, CSP is designed to be effective without extensive adjustments. We introduced possible extensions, but our findings suggest that the marginal performance gains from these variations wouldn’t change CSP’s practical value as an efficient baseline. If higher performance is required, more advanced methods are likely preferable. That said, exploring these extensions in future work remains an option for enhancing CSP's adaptability.

---

> ### Author Response · Authors · 2024-11-15
>
> # Responses to Reviewer Questions
>
> ## Tradeoffs Between Accuracy, Training Time, and Memory Usage
> We evaluated AUC-ROC for classification and P@100 for retrieval to highlight CSP’s retrieval strength. CSP does not have separate training and inference stages due to its parameter-free, transductive setup, allowing us to measure time directly (see Table 4). We didn't directly discuss memory complexity and would like to address it in further revisions. Of note is the fact that CSP is inherenly distributable, alleviating some limitations w.r.t. memory complexity. However, in our experiments on confidential data numbering in the tens of billions of nodes, memory usage was manageable even in a non-distributed setting.
>
> ## CSP’s Unique Contribution Compared to Existing Work
> CSP’s contribution lies in its simplicity and efficiency. Unlike many methods that calculate embeddings, CSP avoids this step entirely, making it an attractive option when computational resources are limited. By omitting learnable parameters, CSP also stands out as a non-parametric, versatile baseline that can be quickly applied without extensive preprocessing, training, or tuning. In addition, ability to implement the method in few lines of SQL code makes it attractive for data-oriented enviroments, where the query can be executed directly in the data center.
>
> ## Inclusion of General Hypergraph Neural Networks [b, c] in Experiments
> While we did not include every recent hypergraph neural network variant in our comparisons, CSP’s distinct lack of backpropagation and training offers a marked contrast. Adding each neural variant could certainly provide further insights, but CSP’s unique value as a parameter-free approach is its ability to provide reasonable performance without these computational costs. We would like to emphasize a few key points in regards to the usefulness of CSP:
>
> - CSP is the only optimization-free method compared
> - CSP is the only parameter-free method compared
> - CSP doesn't aim to achieve the best performance
> - CSP is almost always competitive in performance
> - CSP offers the best performance per unit of computational time
>
> We are aware that these points could be explained more clearly in the paper itself and will try to address that.
>
> ## Comparison with Non-Neural Hypergraph Methods on Datasets Lacking Vertex Features
> This is definitely a valid point that we would like to address in the revised version, thank you for the related references!
>
> Thank you for your suggestions, we're looking forward to continuing the discussion.

---

### Official Review · Reviewer_jT67 · 2024-11-01

**Soundness:** 1
**Presentation:** 2
**Contribution:** 1
**Rating:** 3
**Confidence:** 4

**Summary:**

This paper introduces Convolutional Signal Processing (CSP), a simple, nonparametric, and scalable baseline for hypergraphs. It further examines the relationship between CSP and previously established baselines, such as Hypergraph Convolution, Label Propagation, and Naive Bayes. The method is validated on both classification and retrieval tasks.

**Strengths:**

* Paper is well written and easy to follow.

* The proposed method is simple and scalable.

**Weaknesses:**

* The authors observe that their proposed method resembles a simplified form of Hypergraph Convolution, where learnable weights are replaced by identity matrices. This extension is straightforward and lacks novelty. To strengthen the contribution, a clean empirical analysis or theoretical analysis could clarify the advantages of this design choice over standard hypergraph convolutional methods. Such an addition would enhance the novelty of the approach and provide insights into the specific benefits and trade-offs introduced by this simplification.


* Inconsistent evaluation compared to prior work: previous studies [1, 2] typically employ a 50%/25%/25% train/validation/test split, whereas CSP uses a 90%/10% train/test split. The authors should either align their evaluation with a similar evaluation split in [1, 2] or provide a clear justification for their choice of the 90%/10% split. Additionally, it would be beneficial for the authors to specify whether the recommended hyper-parameters in Hypergraph Convolution were used for the baseline methods and detail the specific settings employed for each baseline.


* Retrieval tasks report only P@100; however, it would be more informative to also report NDCG@K (K=1/10/20). The authors should provide a rationale for their choice of P@100, explaining why this metric was selected over others.
CSP can only support binary labels. What are some implications of this limitation for real-world applications? Were any extensions considered to support multi-class problems?


* Missing baseline comparisons with MLP and Label Propagation. Including these baselines might offer additional insights into performance and advantages of CSP.

---
References

[1] Equivariant Hypergraph Diffusion Neural Operators, ICLR 2023

[2] From Hypergraph Energy Functions to Hypergraph Neural Networks, ICML 2023

**Questions:**

See weaknesses.

Typo: Line 157: consists in a  → consists of a

---

> ### Author Response · Authors · 2024-11-16
>
> Thank you for your review, summary and suggestions. We would like to add a few comments regarding the points that you mention:
>
> TL;DR: CSP achieves competitive performance across tasks while offering unparalleled simplicity, being parameter-free and optimization-free, and delivering the best retrieval efficacy (P@100 per execution time) among all methods tested, with an implementation as simple as a few lines of SQL code. In more detail:
>
> ## Novelty of CSP’s Simplified Approach Compared to Hypergraph Convolution
> You are correct that our method resembles a simplified form of Hypergraph convolution. Our empirical results demonstrate that CSP simplifies computation compared to Hypergraph Convolution (HGCN) without significantly compromising performance. As noted in Table 4, CSP is more computationally efficient by several orders of magnitude, making it particularly advantageous for large-scale data or resource-constrained applications. CSP’s simplicity — not requiring learnable parameters or backpropagation — achieves comparable results without the tuning and computational load that HGCN requires. Apart from this empirical evaluation, we also provide a discussion of the relationships CSP holds to established methods. We welcome any suggestions for further empirical validation and/or theoretical analysis
>
> ## Evaluation Split and Parameter Choices
> The linked previous works indeed use a 50/25/25 split, however, in both works, only a classification scenario is considered. In retrieval tasks, where target classes are usually rare, larger training sets help avoid class imbalance and improve representation of infrequent instances, making the 90/10 split more practical than 50/25/25. Moreover, our primary goal is to evaluate CSP’s efficiency in large-scale retrieval, where extensive training data is often available. We also use default parameter settings for baseline methods (as detailed in Section 5.3), ensuring a fair comparison without heavy parameter tuning.
>
> ## Using P@100 in Retrieval and Binary Label Limitation
> We used P@100 as it aligns with real-world pre-filtering applications where retrieval efficiency matters more than exact ranking. Since our goal was to retrieve a fixed number of relevant items (e.g., 100), P@100 provides a practical measure of CSP’s filtering accuracy. While we did not test NDCG@K metrics, we thank you for this suggestion as it could supplement our results for tasks where ranking order within retrieved items is critical.
>
> For multi-class tasks, CSP is limited to binary signal propagation by design. However, a workaround is possible using one-vs-rest classification or one-hot encoding, where each class signal is independently propagated. This approach worked well in our experiments, but we acknowledge that there remains work to explore these scenarios more thoroughly and future CSP adaptations could extend its usability in native multi-class contexts.
>
> ## Inclusion of MLP and Label Propagation as Baselines
> Traditional Label Propagation could indeed serve as additional baseline, but it introduces challenges in hypergraph contexts. Applying traditional Label Propagation requires converting the hypergraph to a simple graph, which introduces ambiguity and potential information loss, however, we are aware of e.g. Henne's work in generalizing LP to hypergraphs. We agree that at least the hypergraph label propagation method should be included in the revised text for comparison.
>
> MLPs, requiring backpropagation and heavier computation, diverge from CSP’s design goals. We prioritized methods that naturally operate in a hypergraph context or are efficient by design, such as HGCN, to better highlight CSP’s unique simplicity and computational efficiency.
>
> In summary, we thank you for your review, and would like to highlight a few key facts:
>
> - CSP is the only optimization-free method compared
> - CSP is the only parameter-free method compared
> - CSP doesn't aim to achieve the best performance
> - CSP is almost always competitive in performance
> - CSP offers the best performance per unit of computational time
>
> We are aware that these points could be explained more clearly in the paper itself and will try to address that. We're looking forward to continuing the discussion.

---

> > ### Comment · Reviewer_jT67 · 2024-11-17
> >
> > I appreciate the authors' response. I have a few follow-up concerns and suggestions for further clarification:
> >
> > 1. **Training Data Requirements:** While I acknowledge that CSP is both optimization-free and parameter-free, its reliance on large training datasets is a significant limitation. This requirement makes CSP less attractive compared to GNN-based methods, which perform well in low-labeled and semi-supervised settings. Could the authors provide results for a comparable setting (e.g., 50/25/25 split from the previous reference [1])? Given CSP’s efficiency, this comparison should be straightforward to include.
> >
> > 2. **Scalability and Dataset Size:** CSP is positioned as an efficient method, but the experiments are limited to small datasets. To better validate its scalability, could the authors evaluate CSP on larger datasets such as ogbn-arXiv or ogbn-papers100M? This would significantly strengthen the claims of efficiency and scalability. Reference: https://ogb.stanford.edu/docs/nodeprop/
> >
> > 3. **Evaluation Metric:** To enhance the robustness of the retrieval experiments, I suggest re-evaluating a subset of the experiments using NDCG@K as the metric. This would provide a more comprehensive understanding of CSP’s performance in ranking tasks.
> >
> > I look forward to these additional results and am open to revisiting my score if these concerns are addressed satisfactorily.

---

> ### Author Response · Authors · 2024-11-20
>
> We appreciate the thoughtful feedback and suggestions. In hindsight, we may not have clearly positioned CSP in the manuscript. To clarify, CSP is best understood as a simple and efficient baseline method.
>
> ## Training Data Requirements:
> We understand the concern about CSP’s reliance on larger training datasets and agree that including a 50/25/25 split experiment could provide additional insights. However, as CSP is primarily designed for large-scale retrieval tasks, its strengths are better demonstrated through the larger dataset evaluation discussed below.
>
> If time allows, we will include the 50/25/25 split analysis in the revised submission to address this concern. Given that CSP uses only structural information and is far simpler than other methods, we expect a similar performance gap as observed in the larger dataset evaluation presented below.
>
> ## Scalability and Dataset Size:
> Dispite our initial concerns, we are very grateful for raising this point, as addressing it provides an excellent opportunity to highlight CSP’s position within the landscape of available methods. The results not only demonstrate CSP’s simplicity and scalability but also underscore the expected performance gap relative to top-performing methods.
>
> To address your request for a scalability evaluation, we applied CSP to the ogbn-papers100M dataset.
>
> * Retrieval Setup: CSP completed retrieval on this dataset using a standard **16GB RAM** laptop. The process took about 2 minutes, where 1.5 minutes was spent on data loading and **30 seconds** retrieval per one class.
> * Classification Setup: Extending CSP to a multi-class classification task was also straightforward (although not presented in the paper), requiring a one-vs-rest approach with softmax aggregation. This classification process took approximately 55 minutes for all classes due to repeated retrievals for each label. While manageable on a laptop, this demonstrates CSP’s limitations in maintaining its computational edge over methods like Naive Bayes in classification setups.
> * Despite this, CSP’s retrieval efficiency remains unparalleled, as demonstrated by its simplicity and execution time. Furthermore, it is implemented in less than 20 lines of Python code using a single library (polars-for sql execution) + numpy for data loading.
>
> Performance Comparison:
> For classification, CSP achieved **60.9% test** and **63.8% validation** accuracy on ogbn-papers100M, which:
>
> * Falls behind state-of-the-art methods like GNNs (e.g., GLEM+GAMLP at 73.5%).
> * Is comparable to traditional baselines like SGC (66.5%) and Node2Vec (58.1%).
> * CSP relies solely on the structural information from the citation graph, without leveraging additional features like node embeddings, which naturally creates some performance gap compared to feature-rich methods.
> * CSP’s true strength lies in its simplicity. Imagine encountering the dataset for the first time and needing a quick performance estimate. Most methods would require setting up computational clusters, installing dependencies, and fine-tuning parameters. In contrast, CSP can be implemented as a single SQL query with no additional libraries or preprocessing. This positions CSP as a direct alternative to trivial baselines, such as predicting the overall most frequent class for all nodes (2.8% accuracy on test set and 3.3% on validation set), rather than competing with sophisticated methods. Despite this simplicity, CSP delivers 60.9% test-accuracy, proving it to be a practical first-choice baseline for exploratory tasks.

---

> > ### Author Response · Authors · 2024-11-20
> >
> > ### Evaluation Metric – NDCG@K:
> > We will incorporate NDCG@K in the revised submission to provide a more robust evaluation of CSP’s ranking performance in retrieval tasks. Our preliminary results are as follows:
> >
> > | dataset           |   CSP 1 layer |   CSP 2 layers |   Naive Bayes |   random |
> > |:------------------|--------------:|---------------:|--------------:|---------:|
> > | CiteSeer-CC       |         0.524 |          0.582 |         0.510 |    0.142 |
> > | Cora-CA           |         0.727 |          0.739 |         0.715 |    0.123 |
> > | Cora-CC           |         0.577 |          0.706 |         0.558 |    0.125 |
> > | DBLP-CA           |         0.869 |          0.868 |         0.951 |    0.151 |
> > | PubMed-CC         |         0.784 |          0.813 |         0.886 |    0.303 |
> > | corona            |         0.553 |          0.452 |         0.453 |    0.183 |
> > | movielens-ratings |         0.282 |          0.272 |         0.144 |    0.084 |
> > | movielens-tags    |         0.181 |          0.188 |         0.118 |    0.083 |
> >
> > ## Summary - Planned Revisions
> > We are currently working on the following updates:
> >
> > Main changes:
> > * Better emphasize the problem statement and motivation.
> > * Include CSP implementation on the ogbn-papers100M dataset, which highlights CSP’s position within the landscape of available methods.
> > * Add another state-of-the-art parameter-free reference method for performance comparison. As a "direct competitor" to CSP, this method provides a more relevant baseline than fine-tuned graph neural networks. Initial results show CSP remains competitive while maintaining its simplicity.
> >
> > We are considering also other points like including NDCG, ED-HNN that could add additional insight, but are more complex to include.
> >
> > Additionally, we discovered another submitted ICLR paper proposing a related method, offering a slightly different perspective on CSP: https://openreview.net/forum?id=4AuyYxt7A2. While we have not fully explored their approach, it appears CSP is used as a core building block in their method. However, the method itself seems to be significantly more complicated compared to CSP.
> >
> > We are happy to incorporate any additional feedback you may have.

---

> > > ### Comment · Reviewer_jT67 · 2024-11-20
> > >
> > > Thank you for evaluating your method on a large-scale dataset (ogbn-papers100M) and for including the NDCG@K metric. To further enhance transparency and reproducibility, it would be greatly appreciated if you could upload the code and training/inference logs for ogbn-papers100M as part of the supplementary material. I look forward to revisiting my evaluation after verifying these experiments.

---

> ### Author Response · Authors · 2024-11-23
>
> Thank you for your feedback and for revisiting our evaluation.
>
> The code is already available in our [repository](https://anonymous.4open.science/r/CSP-demo-ICLR-2025/papers100M-demo/Readme.md), including:
>
> - A notebook demonstrating the algorithm on a simple graph for clarity.
> - A script for data preprocessing.
> - A script for the CSP algorithm itself.
>
> We also want to apologize for a minor error in our initial evaluation. The true test accuracy for CSP on the ogbn-papers100M dataset is slightly over 0.58, not 0.609 as previously reported. Despite this adjustment, we believe our original claim of CSP's comparable performance with other baselines remains valid. Note that we are able to achieve 60% test accuracy with 2-layers of CSP and alpha 0.8 (details will be available in the revised text).
>
> Thank you again for your helpful suggestions and for giving us the opportunity to improve the transparency and reproducibility of our work. We look forward to your reevaluation.

---

> > ### Comment · Reviewer_jT67 · 2024-11-25
> >
> > I thank the authors for sharing their code, as well as for providing an extended discussion and conducting new experiments. I was able to reproduce the results shared by the authors for the ogbn-papers100M dataset, which I appreciate.
> >
> > That said, I noticed that the implementation closely aligns with the classic Label Propagation (LP) algorithm. The accuracy of CSP might be adversely affected when applied to heterophilic graphs, due to its strong connection to LP. As previously mentioned, CSP appears to be a straightforward extension of Hypergraph Convolution. The work could benefit from more in-depth analysis or theoretical insights to enhance its appeal and impact.
> >
> > Nonetheless, I value the effort the authors have invested in their new experiments and the accompanying implementation. Based on these additions, I have adjusted my score from 1 to 3.

---

> > > ### Author Response · Authors · 2024-11-25
> > >
> > > Thank you for taking the time to reevaluate our work and for reproducing the results on the ogbn-papers100M dataset. We greatly appreciate your effort and constructive feedback.
> > >
> > > Regarding the comparison with Label Propagation (LP):
> > > - **Similar performance:** As noted in Section 4.3, CSP and LP are shown to be equivalent under certain conditions. Therefore, their comparable performance is expected and consistent with our theoretical analysis.
> > > - **Key advantage:** The advantage of CSP over LP lies in its **computational efficiency**, as detailed in Section 4.3:
> > >   > "The matrix multiplication \( H D_e^{-1} H^T \) does not preserve the sparsity of \( H \), which is typical for large hypergraph-datasets. Therefore, the proposed implementation can be significantly more efficient than Equation 7, despite them being mathematically equivalent."
> > >
> > > Regarding heterophilic graphs:
> > > We fully agree that CSP is not suited for heterophilic graphs, and this aligns with its intended purpose. CSP is not designed to outperform advanced, fine-tuned methods on established datasets. Instead, its strength lies in providing **first insights on unexplored, raw real-world data** and enabling **fast iterations** thanks to its simplicity and scalability, as it does not require any specific libraries, can be executed using simple SQL queries, and supports scalability through its inherent ability to be easily distributed across multiple machines. This unique capability addresses a gap that no state-of-the-art method currently fills and constitutes CSP’s major contribution.
> > >
> > > Thank you again for recognizing the effort we invested in our revisions. We are grateful for the opportunity to clarify CSP’s position and contributions further.

---

> > > > ### Comment · Reviewer_jT67 · 2024-11-25
> > > >
> > > > I acknowledge the authors' response and appreciate their efforts. After revisiting the revised paper and discussion, I find my assessment unchanged and will keep my score.

---

### Official Review · Reviewer_DnbE · 2024-11-03

**Soundness:** 3
**Presentation:** 3
**Contribution:** 2
**Rating:** 1
**Confidence:** 5

**Summary:**

This paper extends classic Zhu Gharmani (2003) label propagation to hyper-graphs in a fairly straightforward manner.  Experiments are performed on the quality of the proposed algorithm on some small (and contrieved) hypergraph datasets.  Although they don't contain many recent baselines, the proposed method is still beat by baselines frequently (including naive bayes).

**Strengths:**

+ well principled extension of classic idea
+ clearly written

**Weaknesses:**

- experimental results are on simple datasets and use very simple baselines.  still the proposed method doesn't seem that compelling
- the hypergraph formalization seems excessively complicated for simple recommendation problems studied
- zhu and gharmani is so old that it has doubtlessly been extended to hyper-graphs;  there absolutely must be more related work in this area

**Questions:**

See weaknesses.  Unfortunately there realistically isn't much that could convince me of this paper's novelty or merits.

The related work, discussion, and baselines should focus more on the existing work on LP and hypergraphs (Henne 2015, Villian, etc).  Many more papers in this vein are not compared with, including:

- Hypergraph Propagation and Community Selection for Objects Retrieval (2021)
- Hypergraph Label Propagation Network (2020)
- Wasserstein Soft Label Propagation on Hypergraphs (2018)

.... and doubtlessly many more.

---

> ### Author Response · Authors · 2024-11-15
>
> Thank you for your review, summary and suggestions. We would like to add a few comments regarding the points that you mention:
>
> ## Performance Compared to other baselines
> We would like to address the choice of datasets and baselines in our work. We used a combination of established datasets as used in prior art (e.g. [a]) together with additional datasets to underline the efficiency of CSP (Most particularly the movie-RA and movie-TA) datasets. We welcome suggestions for additional datasets that could significantly add to the comprehensiveness of CSP's experimental evaluation. We have used a mixture of very simple baselines (such as Naive Bayes) which are relevant due to the positioning of CSP as a simple baseline itself. Additionally, we have included a representative of State-of-the-art neural methods for learning on hypergraphs, HGCN. We believe a complete set of comparisons to other complex models would provide limited additional insights given CSP’s intended use as an efficient, non-parametric, optimization-free benchmark. As to the advantages of CSP, we would like to highlight a few key facts:
>
> - CSP is the only optimization-free method compared
> - CSP is the only parameter-free method compared
> - CSP doesn't aim to achieve the best performance
> - CSP is almost always competitive in performance
> - CSP offers the best performance per unit of computational time
>
> We are aware that these points could be explained more clearly in the paper itself and will try to address that.
>
> ## Hypergraph Formalization
> The hypergraph formalization may appear complex, but it is essential for establishing connections between CSP and well-established methods like Naive Bayes, HGCN, and classic Label Propagation. This formalization also allows CSP to be directly applicable across various domains, including recommendation and classification. By setting up CSP within a hypergraph framework, we unify different learning tasks and enable direct comparisons, which helps highlight CSP’s versatility.
>
> ## Related Work and Existing Extensions
> We appreciate the reviewer’s feedback on related work and will review the additional suggested papers thoroughly. Our analysis does note Henne (2015), and we did not find other approaches in the literature that achieve CSP’s simplicity and practical efficiency (e.g., parameter-free, optimization-free, SQL-compatible). Although various methods exist for label propagation on hypergraphs, CSP’s novelty lies in its minimalistic implementation, balancing efficiency with competitive performance across tasks. We welcome suggestions for papers that align with the design goals of CSP.
>
> We would also like to comment on the summary, which in our view does not fairly represent the presented work. We evaluated our algorithm on standard hypergraph datasets (see e.g. [a]) as well as some bigger hypergraphs such as the movie-RA and movie-TA datasets. CSP does not consistently outperform Naive Bayes in classification tasks, except for the movie-RA and movie-TA datasets. However, the situation reverses in retrieval tasks, where CSP outperforms Naive Bayes in 5 out of 7 setups (see Table 3). This difference between scenarios is expected as Naive Bayes is utilizing prior information in contrast to CSP, which is discussed in Section 4.5 (as well as lines 472-475 in the experimental section). Nonetheless, superior performance is not one of the claims of our work, we aim to emphasize comparable performance with sigificantly lower complexity. Thank you for your suggestions, we're looking forward to continuing the discussion.
>
> [a] You are AllSet: A Multiset Function Framework for Hypergraph Neural Networks, In ICLR'22

---

> > ### Comment · Reviewer_DnbE · 2024-11-26
> >
> > I acknowledge the author's rebuttal (and replies to other reviewers).
> >
> > Unfortunately the authors have purposely chosen not to compare with recent work in the field (e.g. VilLain, references from Reviewer 8FB5, etc).
> >
> > Reviewer jT67's response also highlights my novelty concerns.
> >
> > > I noticed that the implementation closely aligns with the classic Label Propagation (LP) algorithm. ... As previously mentioned, CSP appears to be a straightforward extension of Hypergraph Convolution.
> >
> > Given both significant concerns remain unaddressed, I am keeping my review.

---

### Official Review · Reviewer_u9f8 · 2024-11-04

**Soundness:** 2
**Presentation:** 3
**Contribution:** 2
**Rating:** 5
**Confidence:** 4

**Summary:**

In this paper, the authors introduce Convolutional Signal Propagation (CSP), which is a simple and efficient algorithm for learning on large-scale structured data represented as hypergraphs or bipartite graphs. CSP is formulated as an iterative averaging process that propagates signals over hypergraph structure. Authors also analyse the proposed CSP, establishing connections between CSP and label propagation, niave bayes, and one-layer HNN. Experimental results for the tasks of node classification, and retrieval tasks are produced which show that CSP maintains very low computational complexity while achieving decent success. Authors present CSP as a important first choice baseline for learning on hypergraphs and large-scale structured data.

**Strengths:**

1. The CSP algorithm is simple and can be implemented with just a few lines of code, making it computationally efficient.
2. The paper evaluates CSP on a diverse set of real world datasets related to various domains, which show the broad applicability of the proposed method.
3. Despite simplicity, the empirical results show that CSP achieves good performance compared to more complex baseline methods on tasks node classification, and retrieval.
4. Authors also discussed several possible extensions for the basic CSP method.

**Weaknesses:**

1. The proposed CSP is simple, but it does not show any advantages over a simpler Naive-bayes classifier. For node classification task, Naive Bayes is better suited as a baseline than CSP.
2. A simpler baseline is more beneficial on large scale datasets, because running a complex algorithm on small datasets is not that expensive. Among the datasets considered for evaluation DBLP, movie-RA and movie-TA are large-scale. CSP does not achieve better results on large scale datasets for both classification and retrieval tasks. CSP only achieves better results on small scale datasets, this raises the question of whether CSP is needed at all while a simpler Naive Bayes might just do the job.
3. Authors also do not compare with simpler GNNs, which I think should be added in the evaluation.
4. Authors discuss about various possible extensions to the basic CSP, but are not included in experimentation. I do not completely understand the reasoning behind this, but including them in experimental evaluation will strengthen the paper.

**Questions:**

I have addressed my concerns in the weakness section, I do not have any specific questions.

---

> ### Author Response · Authors · 2024-11-14
>
> Thank you for your review, summary and suggestions. We would like to add a few comments regarding the points that you mention:
>
> TL;DR: CSP is simpler than Naive Bayes. CSP outperforms Naive Bayes on the movie-RA and movie-TA datasets. In more detail:
>
> ## Comparison with the Naive Bayes classifier
> CSP is actually even simpler than the Naive Bayes classifier, both conceptually as well as when measured by computational complexity and cost (see Table 4). As you note, CSP does not consistently outperform Naive Bayes in classification tasks, except for the movie-RA and movie-TA datasets. However, the situation reverses in retrieval tasks, where CSP outperforms Naive Bayes in 5 out of 7 setups (see Table 3). This difference between scenarios is expected as Naive Bayes is utilizing prior information in contrast to CSP, which is discussed in Section 4.5 (as well as lines 472-475 in the experimental section). Naive Bayes requires both positive and negative class fitting, adding to its complexity, while CSP only propagates positive signals — making it computationally lighter while still performing comparably in many tasks.
>
> ## Effectiveness on Large Datasets
> We would like to note that CSP actually achieves better results than Naive Bayes on large datasets like movies-RA and movies-TA for both classification and retrieval tasks, sometimes by a significant margin (see Tables 2 and 3). While Naive Bayes performs better on the DBLP dataset (where it even beats HGCN in the retrieval setting), CSP’s primary advantage is its efficiency across datasets — offering competitive performance while requiring fewer computations (see Table 4). Combined with the relative simplicity of CSP even when compared to Naive Bayes, it is still an attractive baseline algorithm.
>
> ## Exclusion of Simpler GNNs
> GNNs do not naturally operate on hypergraphs, so converting a hypergraph to a unipartite graph introduces complexity, ambiguity and an additional processing step, which could complicate comparisons. Given that HGCN directly applies to hypergraphs, it provides a more relevant benchmark. We believe that HGCN's structure captures the benefits of simpler GNNs applied to hypergraphs, which better aligns with CSP’s design focus. At the same time, we welcome suggestions for simple models that operate on hypergraphs and could serve as additional comparisons.
>
> ## Omission of CSP Variants in Experiments
> Our main contribution and message is the comparative performance of CSP despite its radical simplicity, achieving 'good enough' performance as a low-cost baseline. While we briefly discuss extensions, our experiments focus on demonstrating CSP’s core value. We would like to study and experimentally verify the extensions more in future work, however, in this paper, we focused on our primary message: that CSP is a highly efficient, simple alternative that performs comparably with more complex methods.
>
> In summary, we thank you for your review, and would like to highlight a few key facts:
>
> - CSP is the only optimization-free method compared
> - CSP is the only parameter-free method compared
> - CSP doesn't aim to achieve the best performance
> - CSP is almost always competitive in performance
> - CSP offers the best performance per unit of computational time
>
> We are aware that these points could be explained more clearly in the paper itself and will try to address that. We're looking forward to continuing the discussion.

---

> > ### Comment · Reviewer_u9f8 · 2024-11-26
> > **Thank you for the Response**
> >
> > I thank the reviewers for the responses. I have read through all the other reviewers' comments and responses of authors, my score remains the same.

---

### Meta-Review · Area_Chair_Qrgf · 2024-12-21

**Metareview:**

This paper proposes Convolutional Signal Propagation (CSP), a simple and efficient algorithm for learning on large-scale structured data represented as hypergraphs or bipartite graphs. While the reviewers acknowledge CSP's scalability, low computational complexity, and decent performance on node classification and retrieval tasks, they raise significant concerns about its conceptual contribution, experimental evaluation, and competitiveness.

The reviewers note that CSP is essentially a straightforward extension of classic label propagation methods to hypergraphs, with one reviewer characterizing it as a trivial extension of Zhu and Ghahramani's (2003) work. This lack of novelty and substantial conceptual contribution to graph machine learning is a major concern. Furthermore, the experimental evaluation is limited by the use of small and contrived datasets, which may not accurately reflect real-world scenarios. The authors were too dismissive of reviewers valid concerns during rebuttal.

Moreover, CSP's performance is frequently surpassed by simple baselines, including Naive Bayes, which raises questions about its competitiveness and usefulness as a baseline for learning on hypergraphs. While the authors compare CSP to well-known methods such as label propagation, Naive Bayes, and Hypergraph Convolutional Networks, the results do not demonstrate a clear advantage of using CSP.

**Additional Comments On Reviewer Discussion:**

Informative reviews, but authors did not take the feedback seriously enough.

---

### Decision · Program_Chairs · 2025-01-22

Reject